# Rapid Response of Nitrogen Cycling Gene Transcription to Labile Carbon Amendments in a Soil Microbial Community

Peter F. Chuckran,[a,b] Viacheslav Fofanov,[c,d] Bruce A. Hungate,[a,b] Ember M. Morrissey,[e] Egbert Schwartz,[a,b] Jeth Walkup,[e] Paul Dijkstra[a,b]

aCenter for Ecosystem Science and Society (ECOSS), Northern Arizona University, Flagstaff, Arizona, USA
bDepartment of Biological Sciences, Northern Arizona University, Flagstaff, Arizona, USA
cPathogen and Microbiome Institute, Northern Arizona University, Flagstaff, Arizona, USA
dSchool of Informatics, Computing and Cyber Systems, Northern Arizona University, Flagstaff, Arizona, USA
eDivision of Plant and Soil Sciences, West Virginia University, Morgantown, West Virginia, USA

**ABSTRACT**   Episodic inputs of labile carbon (C) to soil can rapidly stimulate nitrogen (N) immobilization by soil microorganisms. However, the transcriptional patterns that underlie this process remain unclear. In order to better understand the regulation of N cycling in soil microbial communities, we conducted a 48-h laboratory incubation with agricultural soil where we stimulated the uptake of inorganic N by amending the soil with glucose. We analyzed the metagenome and metatranscriptome of the microbial communities at four time points that corresponded with changes in N availability. The relative abundances of genes remained largely unchanged throughout the incubation. In contrast, glucose addition rapidly increased the transcription of genes encoding ammonium and nitrate transporters, enzymes responsible for N assimilation into biomass, and genes associated with the N regulatory network. This upregulation coincided with an increase in transcripts associated with glucose breakdown and oxoglutarate production, demonstrating a connection between C and N metabolism. When concentrations of ammonium were low, we observed a transient upregulation of genes associated with the nitrogen-fixing enzyme nitrogenase. Transcripts for nitrification and denitrification were downregulated throughout the incubation, suggesting that dissimilatory transformations of N may be suppressed in response to labile C inputs in these soils. These results demonstrate that soil microbial communities can respond rapidly to changes in C availability by drastically altering the transcription of N cycling genes.

**IMPORTANCE** A large portion of activity in soil microbial communities occurs in short time frames in response to an increase in C availability, affecting the biogeochemical cycling of nitrogen. These changes are of particular importance as nitrogen represents both a limiting nutrient for terrestrial plants as well as a potential pollutant. However, we lack a full understanding of the short-term effects of labile carbon inputs on the metabolism of microbes living in soil. Here, we found that soil microbial communities responded to labile carbon addition by rapidly transcribing genes encoding proteins and enzymes responsible for inorganic nitrogen acquisition, including nitrogen fixation. This work demonstrates that soil microbial communities respond within hours to carbon inputs through altered gene expression. These insights are essential for an improved understanding of the microbial processes governing soil organic matter production, decomposition, and nutrient cycling in natural and agricultural ecosystems.

**KEYWORDS** carbon metabolism, metagenomics, metatranscriptomics, microbial ecology, nitrogen fixation, nitrogen metabolism, nitrogen regulation, nutrient transport, soil microbiology

Address correspondence to Peter F. Chuckran, pfchuckran@gmail.com.

In a short-term laboratory incubation, it was shown that soil microbial communities responded to glucose by rapidly transcribing genes encoding for proteins and enzymes responsible for inorganic nitrogen acquisition, including nitrogen fixation.

mSystems®

Inorganic nitrogen (N) availability in soil dictates several ecosystem-level processes such as plant growth (1), greenhouse gas emissions in the form of nitrous oxide (2), and eutrophication from runoff (3). The transformation of N by soil microbial communities is directly tied to the pool of bioavailable N in soils (4, 5). Thus, understanding the controls of N metabolism in soil microbes is key to determining, and potentially managing (6), the cycling of N in soils. Although genes and regulatory mechanisms for microbial N cycling processes have long been identified in laboratory studies (7–9), the short-term dynamics and controls of N cycling in complex soil communities remain poorly understood. The availability of shotgun sequencing technologies to analyze microbial functioning in soil communities provides an opportunity to enhance our understanding of microbially mediated soil N cycling.

Measuring short-term responses of soil microbial populations to changes in the environment is crucial in understanding the role of microbes in biogeochemical cycling. Most biogeochemical transformations occur during short periods of intense microbial activity, when the active fraction of microbes may be up to 20 times higher than that in bulk soil (10). This stimulation is often the result of a localized increase in nutrient concentrations, such as in the rhizosphere or an area of fresh organic matter decomposition. Despite the importance of these "hot moments," only a few studies (11, 12) have tracked changes in N cycling gene transcription in soils.

Notably, the short-term (hours to days) transcriptional response of N cycling genes to labile C inputs has yet to be determined. Microbial communities experience sudden changes in C and N availability associated with plant root exudation (13), trophic interactions (14, 15), and litter leachate (16). Since soil microbes are typically limited by labile C and energy (17–19), the addition of a C-rich substrate is expected to stimulate growth and activity (20), increasing the demand for N (21). Whether N is derived from the uptake of organic N present in the substrate or mineral N available in the soil depends largely on the C-to-N (C:N) ratio of the substrate (22). For example, in a study by Yang et al. (23), soil microbial communities assimilated organic N during the mineralization of added glycine, but in the presence of glucose, the mineralization of glycine was initially suppressed, and ammonium served as the main source of N. Simple sugars such as glucose have accordingly been shown to influence protease activity (24). The metabolic pathways for N immobilization have been well characterized *in vitro* (25). A majority of N assimilation into biomass occurs through the conversion of $NH_4^+$ into the amino acids glutamine and glutamate, which are used as sources of N for all other amino acids. Under low-to-moderate intracellular concentrations of $NH_4^+$, the enzymes glutamine synthetase (GS) (encoded by *glnA*) and glutamate synthase (GOGAT) (*gltS*) convert $NH_4^+$ to glutamate in a two-step reaction referred to as the GS-GOGAT pathway (26). Under high concentrations of $NH_4^+$, the enzyme glutamate dehydrogenase (GDH) (*gudB* and *gdhA*) converts $NH_4^+$ directly to glutamate in a one-step reversible reaction (27).

Since both the GS-GOGAT pathway and GDH require N as $NH_4^+$, other forms of inorganic N must be converted to ammonium before conversion into biomass. In the case of nitrate and nitrite, the reduction to ammonium occurs through either assimilatory nitrate reduction or, under anoxic conditions, dissimilatory nitrate reduction to ammonium (DNRA) (see Table S1 in the supplemental material) (28). The conversion of atmospheric $N_2$ to ammonium by diazotrophs is catalyzed by the enzyme nitrogenase (*nifDHK*) (29).

The mechanisms regulating N uptake in response to C have been extensively studied *in vitro* (8, 25). The complex regulatory network includes a specialized sigma factor ($\sigma^{54}$; *rpoN*), three transcriptional regulators, and a phosphorylation cascade comprised of postmodification enzymes, PII proteins, and a two-component regulator (30). The activity of many of the enzymes and proteins in the phosphorylation cascade is tightly controlled by cellular concentrations of glutamine and oxoglutarate (31). Since the concentration of oxoglutarate is impacted by the activity of the tricarboxylic acid (TCA) cycle, the regulation of N cycling is directly tied to C metabolism (32).

Carbon substrate addition is also thought to influence dissimilatory N cycling processes such as nitrification and denitrification. In nitrification, ammonia is oxidized to nitrite and then nitrate. Often, the steps of this process occur in different organisms (33); however, complete ammonia oxidizers have also been described (34, 35). In denitrification, nitrate is reduced to nitrite, nitric oxide, and then nitrous oxide and $N_2$. Nitrification and denitrification, beyond their ability to draw from the pools of ammonium and nitrate, also represent important avenues of inorganic N loss from soils via nitrate leaching and the release of $N_2$ and nitrous oxide, a potent greenhouse gas (36). The addition of glucose is expected to have both positive and negative effects on nitrification. Rates of autotrophic nitrification tend to decrease as heterotrophs outcompete autotrophic nitrifiers for ammonium (37), but rates of heterotrophic nitrification may increase after labile C inputs (38). Denitrification is more directly influenced by C availability and quality (39), and the abundance of mRNA transcripts associated with denitrification was stimulated with the addition of glucose in anoxic soil microcosms (40).

Despite our knowledge of the mechanisms and controls of N cycling and N metabolism, we do not yet fully understand how these genes are regulated within complex soil microbial communities. Metatranscriptomics allows us to capture the transcriptional profile of a microbial community, providing insight into the potential activity of a community at a given moment in time (41–43). Many studies utilizing this technique have focused on the influence of ecosystem-level characteristics/properties on transcription, such as land use, aboveground cover, seasonality, and climate (44–49). Although these studies contribute greatly to our understanding of community gene transcription, there is an additional need to study the dynamic short-term responses of microbial communities to changes in C and N availability (50).

In order to fill this knowledge gap, we conducted a soil incubation study where we induced rapid immobilization of inorganic N by adding glucose. We selected glucose as it is a form of labile C commonly found in soils (51) and has been widely used to alleviate C limitation in soil microbial communities as a means to study growth (52, 53) and metabolic activity (50). We analyzed metagenomes and metatranscriptomes of the soil microbial community using high-throughput shotgun sequencing to identify the response of N cycling genes over a 48-h period. We hypothesized that the abundance of N cycling genes in the metagenomes would not significantly change throughout the course of the 48-h incubation but that changes in activity would be immediately detected in the metatranscriptomes. We further hypothesized that there would be an upregulation of genes associated with inorganic N transport, N assimilation into biomass, and N metabolism regulation in response to labile C inputs and that the abundance of these transcripts would track the concentrations of inorganic N. This work provides an in-depth look at the short-term transcriptional response of soil microbial communities during a central biogeochemical process in soils.

## RESULTS

**Biogeochemical measurements.** The concentration of $NO_3^-$ decreased in the 24 h after glucose addition and remained low for the remainder of the incubation (Fig. 1A). The concentration of $NH_4^+$ also decreased during the first 24 h of the incubation and increased thereafter (Fig. 1B). Rates of $CO_2$ production increased from 4 to 16 h and then decreased from 28 to 48 h in response to glucose (Fig. 1C). We found that the addition of water only slightly influenced $CO_2$ production (see Fig. S1 in the supplemental material), indicating that the majority of the stimulation was due to the addition of labile C. $K_2SO_4$-extractable organic carbon decreased for the first 20 h and plateaued thereafter (Fig. 1D). Based on these biogeochemical measurements, we selected 4 time points (0 h [$t_0$], $t_8$, $t_{24}$, and $t_{48}$) from which we extracted DNA and RNA. These time points captured distinct phases of C and N availability that enabled us to test our hypotheses.

Microbial biomass C (MBC) moderately decreased throughout the incubation (Fig. S2A), and microbial biomass N (MBN) remained constant (Fig. S2B). Bacteria may exhibit some

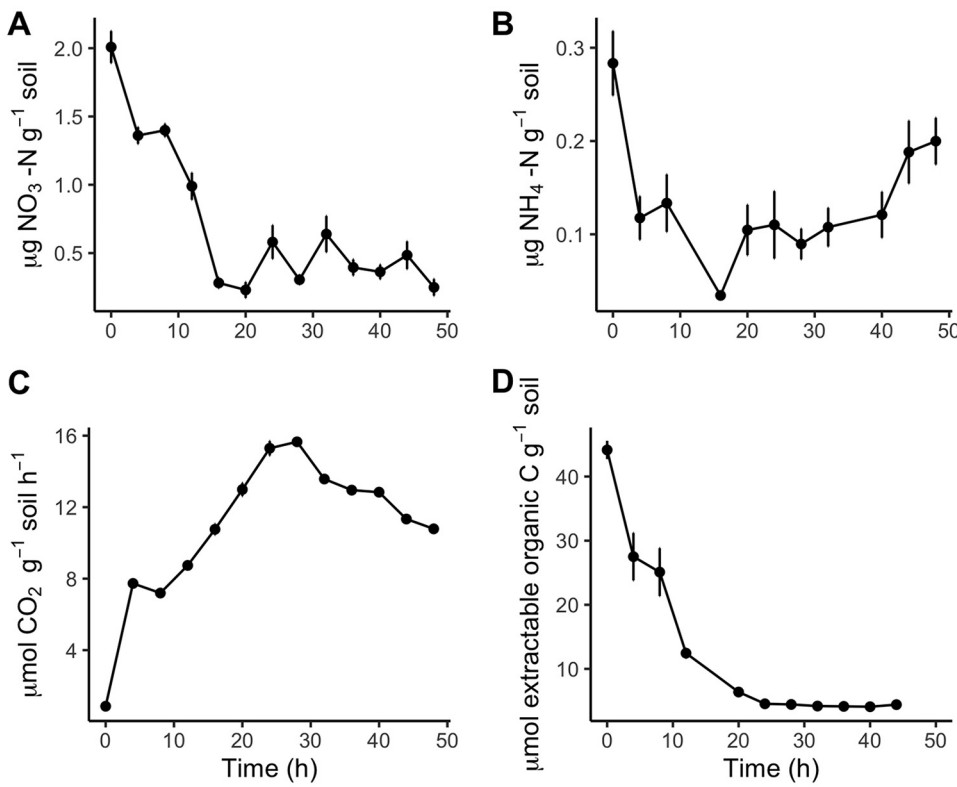

**FIG 1** Mean concentrations ($\pm$ standard errors [SE]) of nitrate (A) and ammonium (B) and rates of carbon dioxide production (C) and $K_2SO_4$-extractable C (D) as a function of time after glucose amendments.

stoichiometric plasticity in response to nutrient inputs (54); however, a decrease in the biomass C:N ratio in response to C inputs is counterintuitive. Since the method of microbial biomass extraction used involves two extractions on the same sample (one before and one after fumigation), incomplete extraction of the added glucose in the first extraction could result in an artificially high estimate of biomass C. We believe that it is far more likely that microbial biomass and stoichiometry did not change and that changes in the estimated MBC are likely the result of unextracted glucose remaining from the initial $K_2SO_4$ extraction.

**Metagenomic and metatranscriptomic assembly and annotation.** Out of 16 soil samples from which DNA and RNA were extracted, 12 were successfully sequenced and assembled for metagenomic analysis, and all 16 were successfully sequenced and assembled for metatranscriptomic analysis. For the metagenomes, the proportion of genes successfully annotated against the Kyoto Encyclopedia of Genes and Genomes (KEGG) database varied from 23.4% to 25.6% of all genes per sample. Of the 6,876 functional KEGG orthologs identified in the metagenome analysis, 671 genes were present in higher abundances, while 332 were present in lower abundances (false discovery rate [FDR] < 0.01), after the addition of glucose. Glucose caused a shift in the relative abundance of functional genes (permutational multivariate analysis of variance [PERMANOVA], $F_{3,11}$ = 3.24 [$P$ < 0.01]) (Fig. 2, top). The genes that were most different in gene abundance relative to $t_0$ varied for each time point (similarity percentage [SIMPER] analysis) (Table S3A), and not one of these genes was directly related to N uptake. Among these were the subunits of RNA polymerase *rpoB* and *rpoC*, which were present in slightly lower abundances at $t_8$ (log$_2$ fold change [LFC], $-0.1$; FDR > 0.1), and the regulatory gene for the *lac* operon, *lacI*, which was present in a higher abundance at $t_{24}$ and $t_{48}$ (LFC, 0.7; FDR < 0.01). The largest changes were found at $t_{24}$ for low-abundance spore germination proteins (Table S3B), specifically *gerKC* (KEGG number K06297) and *yfkQ* (KEGG number K06307), which were 8.8 and 7.4 log$_2$-fold more abundant than at $t_0$.

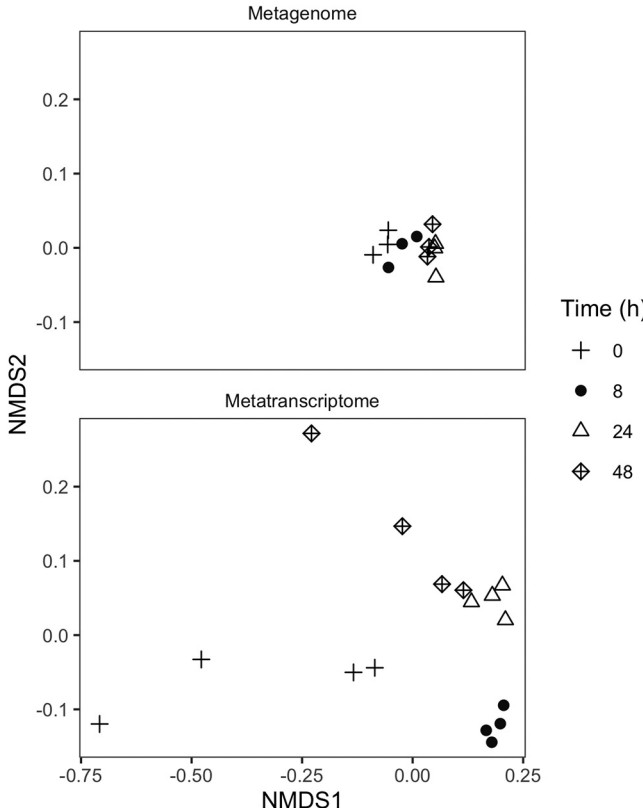

**FIG 2** Nonmetric multidimensional scaling (NMDS) using Bray-Curtis distance of normalized KEGG annotation abundances for metagenomes (top) and metatranscriptomes (bottom) 0, 8, 24, and 48 h after the addition of glucose.

The proportion of transcripts successfully annotated against the KEGG database varied between 12.6% and 32% of all transcripts in a metatranscriptome. Transcripts for 5,448 functional genes were identified, of which 1,141 were increased and 855 were decreased in response to glucose. PERMANOVA indicated significant shifts in the abundance of transcripts between time points ($F_{3,15}$ = 8.07 [$P < 0.01$]) (Fig. 2, bottom). Transcripts encoding *amt* and *glnA* contributed the most to the dissimilarity with $t_0$ (SIMPER analysis); combined, they explained 1% of the differences at $t_8$, 1% of the differences at $t_{24}$, and 0.9% of the differences at $t_{48}$.

**Gene and transcript abundances of nitrogen cycling processes.** The abundance of N cycling genes was generally stable over time (Fig. 3A), with changes in gene abundance often being several orders of magnitude smaller than changes in transcript abundances. For metatranscriptomes, many genes associated with N uptake were highly upregulated in response to glucose (Fig. 3). The expression levels of genes encoding the GS-GOGAT pathway (GS, *glnA*; GOGAT, *gltS*, *gltD*, and *gltB*) were consistently upregulated after glucose addition (FDR < 0.01), peaking at 8 h (Fig. 3B; Table S2). We did not find a similar trend for transcripts associated with glutamate dehydrogenase (GDH) (*gudB* and *gdhA*). Instead, we found variable increases and decreases in the expression levels of these genes, which corresponded to different classes of GDH enzymes (Fig. 3B; Table S2). In prokaryotes, GDH often uses NADH (EC 1.4.1.2) and NADPH (EC 1.4.1.4) as cofactors, while GDH in eukaryotes can use both [NAD(P)H] (EC 1.4.1.3) (55). The transcription of genes for EC 1.4.1.4 significantly increased early ($t_8$, LFC of 1.542 ± 0.312 and FDR of <0.01), and transcription for EC 1.4.1.2 trended higher later ($t_{48}$, LFC of 2.229 ± 0.884 and FDR of <0.1). The eukaryotic EC 1.4.1.2 gene GDH2 (K15371) was upregulated at $t_{24}$ (LFC of 1.350 ± 0.434 [Table S2] and FDR of <0.01), and EC 1.4.1.3 was slightly downregulated throughout (significantly at $t_8$ [FDR < 0.01]).

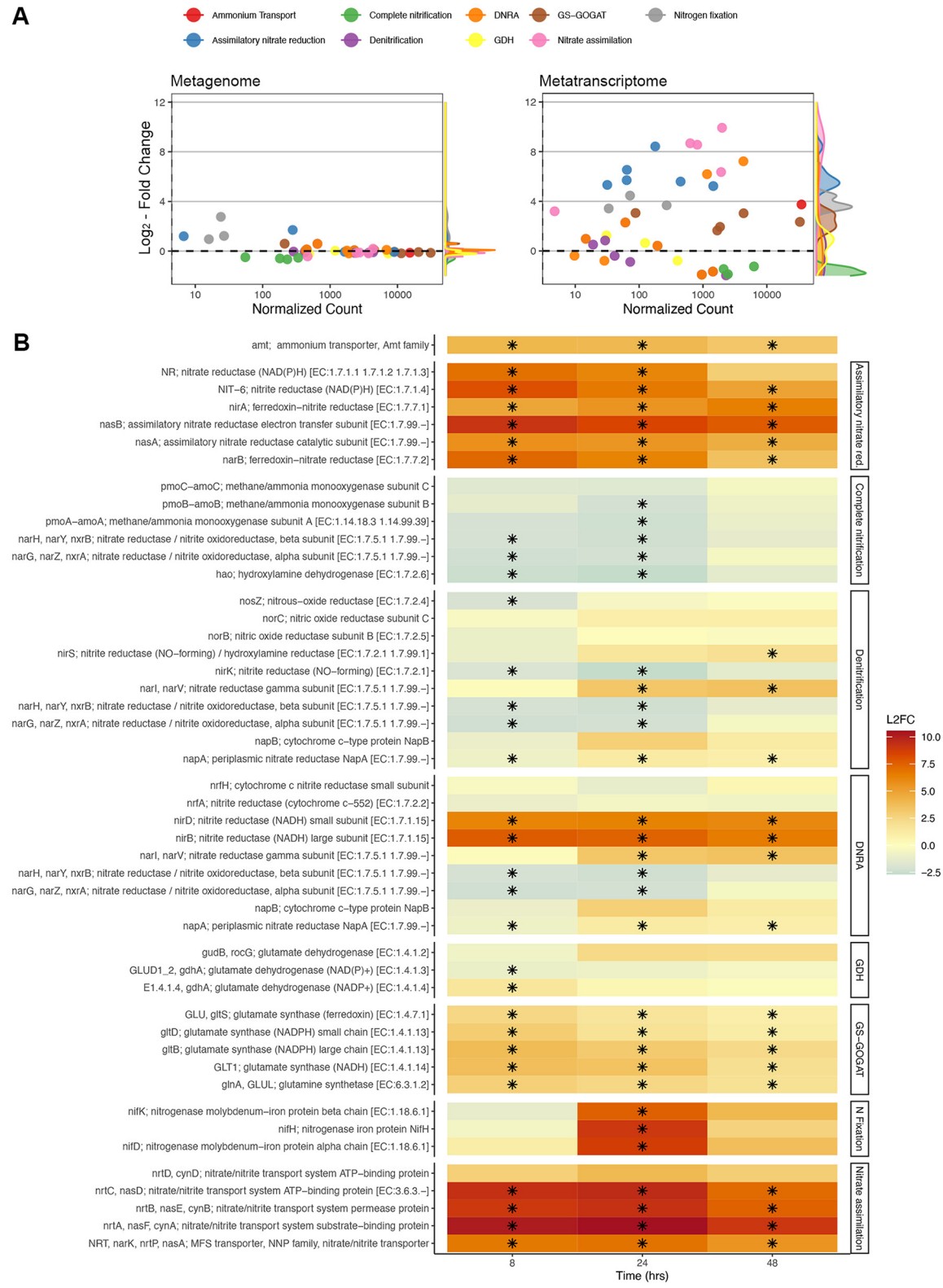

**FIG 3** (A) Log$_2$ fold changes (mean LFCs $\pm$ SE) relative to $t_0$ of normalized gene (left) and transcript (right) abundances versus normalized counts for N cycling genes from glucose-amended soils. LFCs and normalized counts represent the averages between $t_8$, $t_{24}$, and $t_{48}$ for each gene. (B) Log$_2$ fold changes in transcript abundances for genes grouped by biologically relevant reactions and pathways. A black asterisk indicates a significant change relative to $t_0$. MFS, major facilitator superfamily.

The abundance of transcripts encoding the ammonium transporter AmtB (*amt*) was significantly (FDR < 0.01) higher after glucose addition throughout the 48-h incubation (Fig. 3B; Table S2), peaking at $t_8$, where it was 16-fold higher than that at $t_0$ (41,366 transcripts at $t_8$ versus 2,539 at $t_0$). A similar upregulation was found for genes associated with nitrate and nitrite transport across the membrane: 1,500-fold increases compared to $t_0$ (from 2.6 to almost 2,800 transcripts per sample at $t_{24}$) (Fig. 3B).

Genes associated with assimilatory nitrate reduction (Fig. 3; Table S2) were strongly upregulated at $t_8$ and remained upregulated over the 48-h incubation period. In contrast, we found variable responses of genes associated with DNRA. Most genes associated with the dissimilatory reduction of nitrate to nitrite were downregulated or not significantly affected, with a few exceptions. Nitrate reductase gamma subunits (*narI-narV*) were upregulated at $t_{24}$ and $t_{48}$, and the genes *nirB* and *nirD*, which encode the small and large subunits of the cytosolic enzyme nitrite reductase, were significantly (FDR < 0.01) upregulated throughout the incubation (LFC, 6.18 to 7.70) (Fig. 3B). In contrast to these enzymes, the abundances of transcripts that encode a periplasmic cytochrome *c* nitrite reductase (*nrfA* and *nrfH*) did not significantly change in response to C amendment.

The expression levels of all genes involved in nitrification were downregulated in response to glucose, and a majority of these genes (5 of 6) were significantly (FDR < 0.01) downregulated at some point during the incubation (Fig. 3B). Similarly, the expression levels of most denitrification genes were downregulated throughout the incubation, with the exception of *narI* and *narV*, which encode gamma subunits of nitrate reductase.

Transcripts for three genes that encode subunits of nitrogenase (*nifK*, *nifD*, and *nifH*) were detected, all of which were present at very low abundances at $t_0$, $t_8$, and $t_{48}$. Only at $t_{24}$ did we observe a strong significant (FDR < 0.01) upregulation for all 3 genes, up to 410-fold higher than that at $t_0$ for *nifH* (798 transcripts at $t_{24}$ versus 1 at $t_0$) (Fig. 3B).

We found that the vast majority of N cycling gene transcription could be attributed to bacteria and archaea (Fig. 4). Dissimilatory processes were largely from *Thaumarchaeota* and *Nitrospirae*, while assimilatory processes tended to be represented by *Proteobacteria*, *Actinobacteria*, and *Acidobacteria*. Nitrogen fixation was heavily dominated by *Proteobacteria* (Fig. 4).

**Regulation of N cycling genes.** Generally, transcripts of genes associated with the regulation of N metabolism increased after glucose addition (Fig. 5; Fig. S3). The abundances of uridylyltransferase (UTase; *glnD*) but not adenylyltransferase (ATase; *glnE*), used for postmodification of glutamine synthetase (GS) and regulatory PII proteins, respectively, initially increased at $t_8$ (2.18 ± 0.41 LFC and 4.31 ± 0.36 LFC; FDR < 0.01) (Fig. 5; Fig. S3). UTase (*glnD*) but not ATase (*glnE*) continued to be significantly upregulated at $t_{24}$ (3.79 ± 0.36 LFC) and $t_{48}$ (2.75 ± 0.36 LFC) (Fig. S3). A similar upregulation was noted for the PII proteins GlnB (*glnB*) (LFC, >2.9; FDR < 0.01) (Fig. S3) and GlnK (*glnK*) (LFC, >3.9; FDR < 0.01) (Fig. S3) and the NtrC family genes *glnL* (FDR < 0.01) and *glnG* (FDR < 0.01 at $t_8$ and $t_{24}$) (Fig. S3). No significant changes in transcript abundances were found for the transcriptional regulators *nac* and *lrp*, while *crp* and *rpoN* were slightly downregulated (LFC of less than −1) at $t_8$ and $t_{24}$ (FDR < 0.01) (Fig. 5; Fig. S3).

**C metabolism.** The LFC and the total number of normalized transcripts for processes involved in glucose breakdown (KEGG modules M00001, M00003, M00004, M0008, and M00009) increased from $t_0$ to $t_8$ and $t_{24}$ (Fig. S4) ($P < 0.05$ by Tukey's honestly significant difference [HSD] test). Significant changes in transcript abundances after glucose amendment were found for the Entner-Doudoroff pathway and the TCA cycle, including the enzyme isocitrate dehydrogenase (*icd*), which produces oxoglutarate, a metabolite that directly connects C and N metabolism (Fig. 5; Fig. S4B; data not shown).

## DISCUSSION

Over a period of 48 h after glucose addition, we observed a substantial decrease in $K_2SO_4$-extractable organic C, an increase in the $CO_2$ production rate, and an increase in the abundance of transcripts for genes associated with glucose breakdown. These

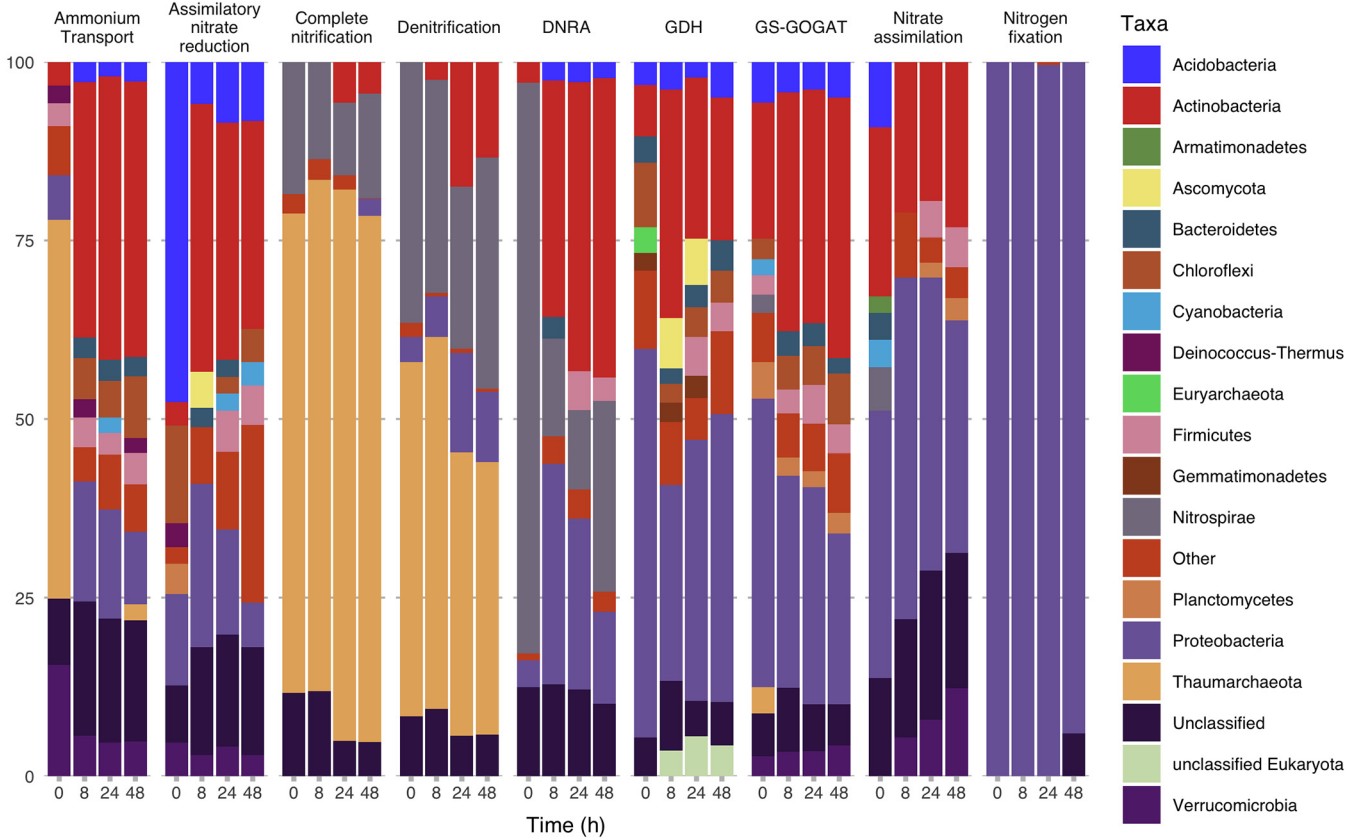

**FIG 4** Relative transcript abundances of major taxa for reactions and pathways of N cycling 0, 8, 24, and 48 h after glucose amendments.

changes coincided with a decrease in inorganic N and an increase in the transcript abundance of genes involved in inorganic N uptake, assimilation, and N metabolism regulation. These results demonstrate that soil microbial communities respond to labile C not only by upregulating genes associated with C metabolism but also by rapidly increasing the transcription of genes responsible for N acquisition. Furthermore, we found that genes for several forms of N acquisition (e.g., N fixation, assimilatory nitrate reduction, and ammonium transport) were differentially transcribed over the 48-h incubation, indicating that changes in multiple microbially mediated N transformations occur within this small temporal window.

**Inorganic N uptake and assimilation.** The GS-GOGAT pathway appeared to be the predominant pathway through which ammonium was assimilated into biomass. The other main avenue of ammonium assimilation into biomass, the enzyme GDH, did not show a similar increase in transcript abundance, and the abundance of GDH transcripts was substantially lower than that of GS-GOGAT. This suggests that GS-GOGAT may be the dominant pathway for the assimilation of inorganic N in soil microbial communities responding to labile C inputs. This finding is consistent with the notion that GDH is most active when the $NH_4^+$ concentration is high and the availability of C is low (27). Assays of soil microbial communities have also shown that GS activity increases in response to higher C-to-N ratios, whereas GDH activity decreases (56). Furthermore, we found that the regulation of GDH transcription appeared to be gene specific, with transcription for EC 1.4.1.4 increasing early and that for EC 1.4.1.2 increasing late. These results nicely follow the concentrations of $NH_4^+$, as NADPH-specific enzymes (EC 1.4.1.4) are generally used for ammonium assimilation (57), whereas NADH-specific enzymes (EC 1.4.1.2) are commonly used for the breakdown of glutamate to ammonium (58). These findings highlight the potential utility of measuring GDH and GS-GOGAT gene transcription for tracking the C and N balance within microbial communities at a given moment in time, which

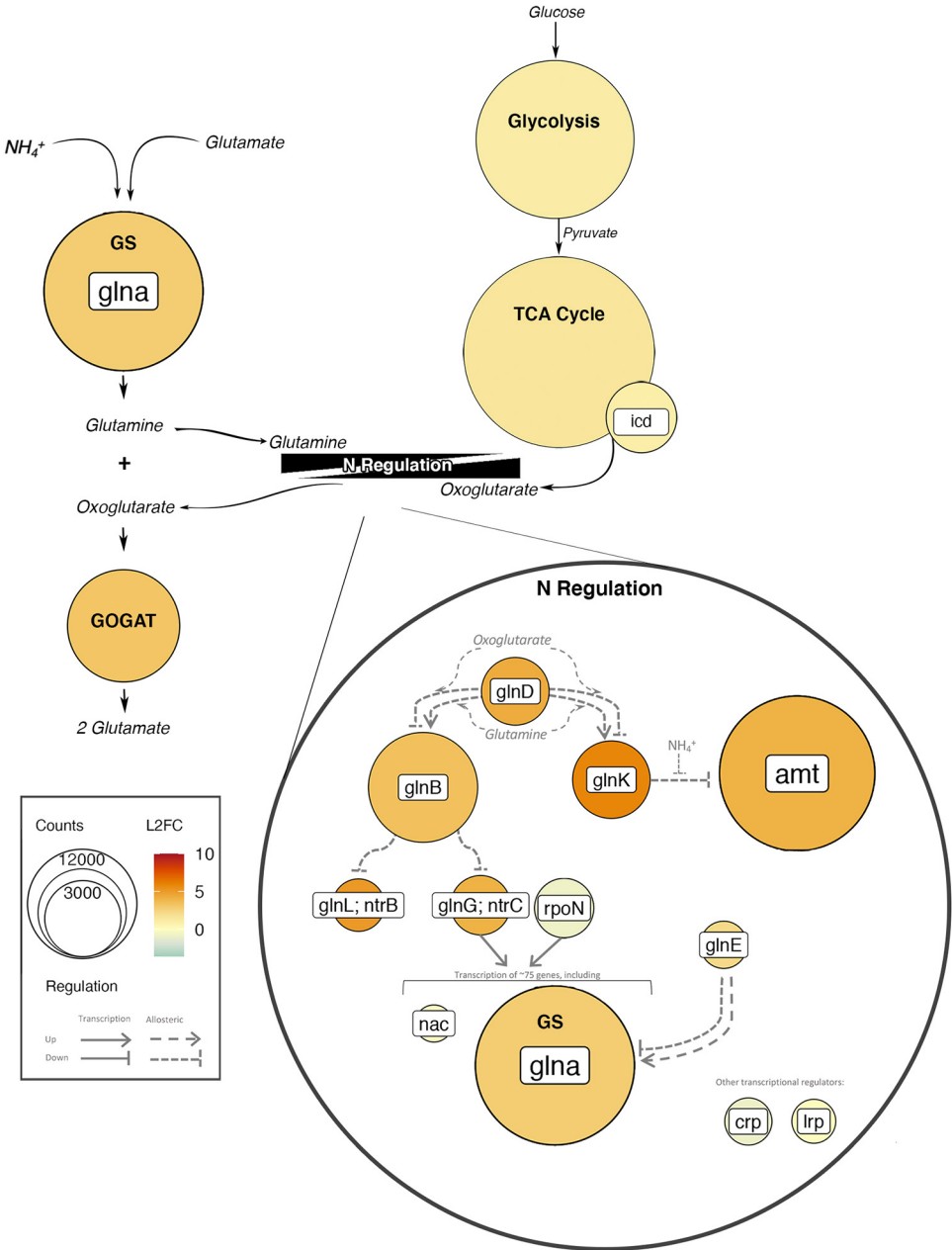

**FIG 5** Abundances and log$_2$ fold changes of transcripts 8 h after glucose addition for C and N metabolism, including glycolysis, the TCA cycle, the N regulatory network, and GS-GOGAT. Color represents log$_2$ fold changes of transcript abundances relative to $t_0$, and size indicates the number of transcripts. Thin black arrows indicate reactants or products of pathways, and gray arrows represent regulatory controls. Gene names are presented in white boxes (for example, *glnA*), whereas pathway or enzyme names are presented in boldface type (for example, GS or glycolysis).

could be a useful approach when, for example, assessing how specific land-use practices influence microbial metabolism and N cycling.

Various mechanisms for transporting inorganic N across the cell membrane were upregulated in response to glucose inputs. Notably, the gene *amtB*, which encodes the ammonium transporter AmtB, was the second most abundant upregulated gene during the incubation (behind *glnA*). Similarly, we observed an upregulation of genes associated with nitrate and nitrite transport (KEGG module M00615) and assimilatory nitrate reduction, which coincided with a precipitous drop in the concentration of NO$_3^-$. Most genes involved in DNRA were not differentially expressed, indicating that

nitrate reduction was primarily occurring under aerobic conditions. Notable exceptions were the genes *nirB* and *nirD*, which encode the cytosolic enzyme nitrite reductase NirBD (59), which has been shown to be active in aerobic soils (60, 61) and may function as the nitrite reductase in assimilatory nitrate reduction (62). Although the upregulation of N transport genes in response to glucose is certainly not novel (30), these results are the first demonstration of this response in a soil microbial community metatranscriptome. Furthermore, these responses show the short time frames (within 8 h) in which soil microbial communities can respond to changes in C and N availability.

The finding that glucose addition strongly upregulated genes encoding nitrogenase, especially when $NH_4^+$ concentrations were low, is consistent with the idea that nitrogen fixation increases when N concentrations are low (63). N fixation has been shown to be activated by the addition of other limiting nutrients such as carbon or phosphorus (64, 65). We therefore believe that the upregulation of nitrogenase genes is a response to low concentrations of $NH_4^+$ and the availability of labile C. The prompt upregulation, and subsequent downregulation, of nitrogenase genes also suggests that some portion of biological nitrogen fixation occurs rapidly in soils or at the very least that the process is highly sensitive to concentrations of $NH_4^+$.

**Connections between C and N metabolism.** Interestingly, transcripts associated with $NH_4^+$ and $NO_3^-$ transport maintained their high abundances despite concentrations of $NO_3^-$ stabilizing and concentrations of $NH_4$ increasing (24 to 48 h into the incubation). One possible explanation is that the activity of these proteins is dictated through allosteric regulation, which is tightly connected to the activities of both C and N metabolism (Fig. 5). For example, the ammonium transporter AmtB is allosterically inhibited by the PII protein GlnK, which is indirectly controlled by internal concentrations of glutamine, an intermediate of N uptake through GS-GOGAT (Fig. 5), and oxoglutarate, an intermediate of the TCA cycle (Fig. 5) (32, 66). In this way, internal concentrations of metabolites from both C and N metabolism may dictate N uptake.

The transcription of N regulatory genes reflects the importance of intermediate metabolites in regulation. We found that the abundances of transcripts for transcriptional regulators (such as *nac*, *lrp*, and *crp*) and $\sigma^{54}$ were either not affected or slightly reduced (Fig. 5). In contrast, transcripts for genes in the phosphorylation cascade, which links C and N metabolism through intermediate metabolites, were more abundant after the addition of glucose (Fig. 5). The upregulation of the two-component regulatory system NtrB (*glnL* and *ntrB*) and NtrC (*glnG* and *ntrC*) within this cascade is especially noteworthy as this system regulates ~75 genes associated with N acquisition, including glutamine synthetase (Fig. 5) (67).

Since the activity of this regulatory network is tightly controlled by internal concentrations of metabolites (30), it is not possible to determine the activity of many of these proteins through the metatranscriptome alone. However, it is noteworthy that almost all of the genes within this regulatory network were upregulated, even if the encoded protein potentially inhibited N transport or assimilation (e.g., GlnK) (Fig. 5). This broad upregulation of genes in the phosphorylation cascade may be beneficial during C uptake, as it allows the concentration of nutrients and metabolites to control N uptake, thereby ensuring that N uptake matches the supply of C (25, 32).

**Nitrification and denitrification.** Most genes associated with nitrification and denitrification were significantly downregulated. Since nearly all nitrifiers in this soil were autotrophic archaea (68), this finding is consistent with the premise that the addition of glucose reduces the rates of autotrophic nitrification by reducing the amount of available ammonium (37). It is not especially surprising that we did not find an upregulation of denitrification genes, as denitrification is most prevalent in anoxic systems with high availabilities of nitrate.

**Genetic potential versus transcription.** Notably, although we observed a slight shift in the functional composition of our metagenomes, these changes did not track those found in the metatranscriptomes in either magnitude or direction. Changes contributing the most to dissimilarity tended to be slight shifts in highly abundant genes, such as *rpoB*, *rpoC*, and *lacI*. We found interesting differences in the abundances of

spore-forming proteins as nutrient availability declined; however, since many of these proteins were uncommon and present in low abundances, the chance of obtaining a false-positive result is much greater, and we are therefore cautious to draw any conclusions based on these data alone. Changes in gene abundance for most N cycling genes were absent. These results suggest that understanding the response of soil microbial communities to short-term changes in the environment necessitates looking beyond the metagenome, as consequential microbial responses occur through changes in gene expression. This is in line with other studies where the composition of transcripts shifts over hours or days (12, 69), whereas shifts in metagenomic community composition have been shown to occur after weeks or months (70).

Our work represents a preliminary look into the short-term transcriptional response of microbial communities to a change in C availability; however, there are a number of considerations moving forward. More work needs to be done focusing on this response in a variety of soils, as nutrient availability and other soil properties will undoubtedly influence this process. For example, soils high in C and low in N would likely not demonstrate a response similar to the one observed for this agricultural soil. Understanding how ecosystem properties influence the dynamics of transcriptional profiles is therefore necessary for determining short-term microbial contributions to biogeochemical cycling. Furthermore, this work focused on a relatively short time frame; however, whether this increase in transcription persists or influences nutrient cycling on the scale of weeks to months remains to be seen. Finally, future efforts should be made to observe these short-term effects *in situ*. Laboratory incubations are extremely useful for controlling environmental variables and isolating a particular response. However, it is likely that under field conditions, and in the presence of plant roots, factors other than C availability will affect gene expression at the same time and to different degrees, potentially masking the response observed in this short-term laboratory experiment.

**Conclusions.** Our results indicate the strong and rapid upregulation of genes associated with the uptake of inorganic N, assimilatory nitrate and nitrite reduction, the GS-GOGAT pathway, and the regulatory network underlying N cycling. Furthermore, the majority of upregulation occurred in pathways that are largely aerobic and heterotrophic, suggesting that these processes dominate the short-term response to labile C in these soils. Perhaps most importantly, this work highlights the importance of microbial gene transcription in controlling short-term biogeochemical cycling in soils. Within the 48-h incubation, we found that microbially mediated transformations of N were well reflected in the metatranscriptome but not in the metagenome or in microbial biomass. The short-term transcriptional responses of soil microbes may therefore serve an important role in determining how biogeochemical fluxes respond to immediate changes in the environment.

## MATERIALS AND METHODS

**Soil sampling and site description.** Soils were collected in the fall of 2017 from a long-term crop rotation experiment at the West Virginia University Certified Organic Farm near Morgantown, WV (39.647502°N, 79.93691°W; 243.8 to 475.2 m above sea level) (68, 71). Samples were taken from plots subjected to a 4-year conventionally tilled crop cycle consisting of corn, soybean, wheat, and a mix of kale and cowpea. Manure was added every 2 years (during corn and wheat planting), and rye-vetch was added as a winter cover crop before replanting corn in the spring. From each plot, 10 cores at a 0- to 10-cm depth were collected and pooled.

**Laboratory incubation.** Soil samples were shipped on ice to Northern Arizona University in Flagstaff, AZ. Soils from 3 plots were pooled, cleaned of roots and large debris, passed through a 2-mm sieve, and distributed among 64 glass Mason jars (500 ml), generating microcosms containing 30 g of soil each. The soil was preincubated at laboratory temperature (~23°C) for 2 weeks prior to glucose addition.

The microcosms received 1.6 ml of a 0.13 M glucose solution, which added 0.7 mg of glucose C g$^{-1}$ dry soil and raised the moisture content to 60% water-holding capacity. Concentrations of glucose in this range have been demonstrated to stimulate soil microbial communities without creating a detrimental increase in osmotic pressure (52). Moreover, a brief trial incubation was conducted to ensure that this concentration of glucose would stimulate $CO_2$ production. Soils were incubated at laboratory temperature (~23°C) under ambient lighting but never direct sunlight. Every 4 h, over a 48-h period, 5

jars were randomly selected and destructively sampled. From each jar, we measured the headspace $CO_2$ concentration, concentrations of $NO_3^-$ and $NH_4^+$, and microbial biomass. A portion of each sample was immediately frozen using liquid $N_2$ and stored at $-80°C$ for DNA and RNA extraction.

Since the addition of water may stimulate community activity and respiration, especially when starting with very dry soil (72, 73), we measured respiration in a parallel incubation wherein the same volume of water was added without glucose. The headspace $CO_2$ from these jars was measured and compared against the glucose additions in order to determine the overall effect of glucose and water on microbial respiration.

**Biogeochemical measurements and analysis.** To measure soil $NO_3^-$ and $NH_4^+$ concentrations, 8 g of soil from each destructively sampled jar was added to 40 ml of a 1 M KCl solution, shaken for 1 h, and filtered through Whatman no. 1 filter paper. Extracts were analyzed on a SmartChem 200 discrete analyzer (Westco Scientific Instruments, Brookfield, CT, USA). Microbial biomass was measured using an extraction-fumigation-extraction technique (74) consisting of a 0.5 M $K_2SO_4$ extraction step followed by a subsequent $K_2SO_4$ extraction step with the addition of chloroform. The first extraction provided an estimate of the $K_2SO_4$-extractable organic C and N from each sample, while the second extraction provided an estimate of microbial biomass C (MBC) and N (MBN). Concentrations of extractable organic C and N were measured on a TOC-L instrument (Shimadzu Corp., Kyoto, Japan). The concentration of $CO_2$ from the headspace of each microcosm was measured using an LI-6262 $CO_2/H_2O$ analyzer (Li-Cor Industries, Omaha, NE, USA) as described previously by Dijkstra et al. (75).

**DNA and RNA extraction and sequencing.** We extracted DNA and RNA just before ($t_0$) and 8 h ($t_8$), 24 h ($t_{24}$), and 48 h ($t_{48}$) after glucose addition ($n = 4$). DNA and RNA were extracted using the RNeasy PowerSoil total RNA kit (Qiagen) according to the manufacturer's instructions. DNA was separated from RNA using the RNeasy PowerSoil DNA elution kit (Qiagen). RNA samples were treated with an RNase-free DNase set (Qiagen) to remove any DNA. Nucleic acid concentrations were determined with a Qubit fluorometer (Invitrogen, Carlsbad, CA, USA), and purity was assessed with a NanoDrop ND-1000 spectrophotometer (Nanodrop Technologies, Wilmington, DE, USA). High-quality samples were sent to the Joint Genome Institute (JGI) for sequencing (76). Paired-end, 2- by 151-bp libraries were prepared using the Illumina NovaSeq platform (Illumina Inc., San Diego, CA, USA).

**Metagenome and metatranscriptomic analyses.** Metatranscriptomes were assembled by the JGI using MEGAHIT v1.1.2 (78) (parameters "megahit ––k–list 23,43,63,83,103,123 ––continue –o out.megahit"), and metagenomes were assembled using SPAdes version 3.13.0 (79). Assembled metatranscriptomes and metagenomes were uploaded to the Integrated Microbial Genomes and Microbiomes (IMG/M) (80) pipeline for annotation. Full details of the bioinformatics pipeline, as well as SRA accession numbers, can be found in the data release (77). From IMG/M, we retrieved the number of reads for all genes attributed to functional orthologs in the Kyoto Encyclopedia of Genes and Genomes (KEGG) Orthology database (81) as well as taxonomic annotations against the IMG database.

Normalization of KEGG functional annotations was performed using the Bioconductor (82) program DESeq2 (83) in R. DESeq2 uses a negative binomial distribution to normalize read counts and estimates the average $\log_2$ fold change (LFC) between harvests. Significant LFCs for each KEGG functional gene and transcript were determined by both a likelihood ratio test (for overall significance) and a Wald test (for specific contrasts between time points) provided in DESeq2. Significance for both tests was assumed at a false discovery rate (FDR) of $<0.01$. Prior to analysis, genes with fewer than 60 reads summed over all samples were discarded in an effort to reduce the FDR correction and improve the detection of significant LFCs (84).

To assess differences in gene and transcript compositions over time, we performed permutational multivariate analysis of variance (PERMANOVA) on our metagenomes and metatranscriptomes. PERMANOVAs were conducted using Bray-Curtis dissimilarity matrices of the square root-transformed normalized read counts with 999 permutations. A SIMPER analysis was used to determine genes that most strongly influenced differences between harvests. PERMANOVAs and SIMPER analyses were conducted using the vegan package (85) for R.

To assess the response of N metabolism to the addition of glucose, KEGG Orthology identifiers (K numbers) were grouped according to KEGG pathways and modules associated with N cycling (86), and K numbers representing regulatory genes controlling N metabolism were identified (8, 25) (see Table S1 in the supplemental material). The response of C metabolism was determined by grouping K numbers by KEGG modules associated with glucose uptake, specifically the Entner-Doudoroff pathway (KEGG module M00008), the TCA cycle (M00009), the pentose phosphate pathway (M00004), gluconeogenesis (M00003), and glycolysis (M00001). From the TCA cycle, we also determined the response of isocitrate dehydrogenase (*icd*), which produces oxoglutarate, an important metabolite linking C and N metabolism (32). Counts and LFCs for K numbers were then averaged for each module to assess the overall response for each process. Results were visualized using the ggplot2 package (87) in R v3.6.1 (88).

**Data availability.** Raw sequence reads and assembled contigs were uploaded to the JGI genome portal (https://genome.jgi.doe.gov/portal/) under GOLD project identifier Gs0135756. A more detailed description of the sequencing and NCBI Sequence Read Archive (SRA) numbers can be found in the data release (77). Contigs are available through the JGI genome portal, and taxonomic and functional annotations of these contigs are available in the IMG/M database (http://img.jgi.doe.gov) under GOLD project identifier Gs0135756. JGI genome identifiers for each sample, as well as sample metadata, were reported previously by Chuckran et al. (77).

## SUPPLEMENTAL MATERIAL

Supplemental material is available online only.
**FIG S1**, TIF file, 0.1 MB.
**FIG S2**, TIF file, 0.2 MB.
**FIG S3**, TIF file, 0.1 MB.
**FIG S4**, TIF file, 0.6 MB.
**TABLE S1**, PDF file, 0.1 MB.
**TABLE S2**, PDF file, 0.1 MB.
**TABLE S3**, PDF file, 0.03 MB.

## ACKNOWLEDGMENTS

This work was supported by funding from the USDA National Institute of Food and Agriculture Foundational Program (award number 2017-67019-26396), and additional support for P.D. was provided by the U.S. Department of Energy Office of Biological and Environmental Research Genomic Science Program LLNL Microbes Persist Soil Microbiome Scientific Focus Area (award number SCW1632). The work conducted by the U.S. Department of Energy Joint Genome Institute, a DOE Office of Science User Facility, is supported under contract number DE-AC02-05CH11231.

We thank Rebecca Mau, Michaela Hayer, Alicia Purcell, and Ayla Martinez for their assistance with laboratory analyses; Sam Bunkers, Kieston Guidry, and Kiara Nelson for their help downloading and cleaning the data; and Isaac Shaffer for his assistance with the analysis. We also thank the Joint Genome Institute for their work in sequencing and assembly, specifically Marcel Huntemann, Alicia Clum, Brian Foster, Bryce Foster, Simon Roux, Krishnaveni Palaniappan, Neha Varghese, Supratim Mukherjee, T. B. K. Reddy, Chris Daum, Alex Copeland, Natalia N. Ivanova, Nikos C. Kyrpides, Tijana Glavina del Rio, and Emiley A. Eloe-Fadrosh.

We have no competing interests to disclose.

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
