## [Reviewer comments · mSystems]

Rapid response of nitrogen cycling gene transcription to labile carbon amendments in a soil microbial community

Peter Chuckran, Viacheslav Fofanov, Bruce Hungate, Ember Morrissey, Egbert Schwartz, Jeth Walkup, and Paul Dijkstra

Corresponding Author(s): Peter Chuckran, Northern Arizona University

Review Timeline:

Submission Date:	February 11, 2021
Editorial Decision:	March 11, 2021
Revision Received:	March 11, 2021
Accepted:	March 30, 2021

Editor: Ryan McClure

Reviewer(s): The reviewers have opted to remain anonymous.

Transaction Report:

DOI: <https://doi.org/10.1128/mSystems.00161-21>

We are grateful for the comments and suggestions of both reviewers and hope that the editor finds the changes we made in response to their suggestions have significantly improved the paper. Please find our responses in *italicized bold* font below each reviewer comment. The line numbers of each change correspond to the “Marked Up Manuscript” file.

Reviewer #1 (Comments for the Author):

This paper explores the global transcriptional effects of glucose amendment to agricultural soil after a 48 hour incubation at four time points (0, 8, 24, and 48 hours). The results generally concur with expectations that when C limitation is alleviated, soil microbes will ramp up nitrogen scavenging pathways to enable growth. This is not surprising, but it is an important and clear result.

We appreciate the reviewers' time and effort in reviewing our work, and appreciate their assessment concerning the importance of our results.

General Comments:

1. Insufficient detail is provided about the incubations. Were they incubated in the light or the dark? Was there any pre-processing to remove roots or other plant components? What was the O₂ content of the incubations?

To address this, we have added more detail about the incubations in L151, 157-161. Specifically, we include more detail about preprocessing and the laboratory light conditions. We do not have data on the O₂ content of the incubations, but given the duration and ratio of soil:container volume, O₂ limitation was extremely unlikely.

2. The paper claims to focus on microbial changes but there is no mention of how the plant and fungal component was filtered out, either physically or via bioinformatics. How much of each metagenome and metatranscriptome was eukaryotic vs. bacterial vs. archaea nucleic acid? Given that the paper provides no information about the taxonomic identity of any of the genes, is it possible that this is not microbial and is actually plant growth? This needs to be explained in more detail. It would strengthen the paper a great deal to include the taxonomy of the top BLAST hits of the metagenomic genes that recruited the most transcripts.

To address this, we have added details about the removal of plant material (L151-152) and added taxonomic annotation data for the N cycling processes discussed in the manuscript. From these data we show that a vast majority of the activity is microbial, specifically bacterial and archaeal. We mention access to these annotations in L210-214. The annotations of N-cycling genes are now included on lines L327-331 and a new Fig. 4.

3. Biomass C: the decrease in soluble C does not look significant in Figure S1. The biomass C results should be toned down given the large variability in the data at each time point.

We agree with the reviewer that there is substantial variability in the microbial biomass C data and have altered the wording to avoid overinterpretation of these trend (L257-261). Specifically, we make the case that the trend over time, although significant, is likely the result of a caveat in the experimental design and not a real change in microbial biomass. We believe that including these results is important, as a substantial change in microbial biomass would alter the interpretation of our results and is therefore likely of interest to many readers.

3. Please add some discussion of the possible variations between laboratory vs field conditions (temperature, light, bottle effects, etc) and how that may have affected the transcriptional response.

We have added a paragraph at the end of the Discussion which discusses some of the limitations of our approach and future considerations for measuring the response detailed in this manuscript. Included is a discussion of the advantages and disadvantages of the laboratory incubation. L469-472

4. The paper is not sufficiently referenced. The authors should include more references of previous studies that showed from activity/isotopic analyses that nitrogen uptake was stimulated by glucose addition to agricultural soil. This paper may be one of the first metatranscriptomes, but surely many previous studies have explored effects at the activity level from chemical measurements of nutrient uptake, enzyme activity, and/or isotope incorporation, or elemental change in plants? Please cite some of these studies to show how this study connects to previous reports that used different methods and as justification for the hypotheses presented in the introduction. I also note in the specific comments below additional instances if missing citations that require references.

To address this, we have provided more background on how labile carbon inputs may influence nitrogen acquisition (L73-L79). Specifically, we discuss the mechanisms responsible for N immobilization in response to glucose amendments and cite examples where this effect has been demonstrated in past work. We have also added more references to the introduction at L103-104 in response to the specific comment below.

5. The authors should consider binning their metagenomes into MAGs and then recruiting the transcripts to the MAGs to obtain a more complete picture of what was happening in their experiments at taxonomic level changes, eg exactly what taxa were stimulated. It would also be much more helpful to do binning and deposit the bins for the community to more easily access instead of the raw sequences which are not as easily accessible on public databases.

Although we fully agree that the addition of MAGs would be an interesting contribution to this analysis, we are primarily interested in whole community responses as opposed to the activity of individual taxa. We believe that understanding community level responses is valuable, particularly for soil microbial ecology, in linking biogeochemical cycling to microbial activity, and accordingly have made this the focus of this work. An analysis of MAGs would undoubtedly be a compelling story and potentially yield valuable insight into features of

responsive taxa, however we believe that this is beyond the scope of this work and would best be served by a separate analysis dedicated to this topic.

We appreciate the reviewers concerns about access to sequencing information. In order to highlight the availability of contigs, as well as functional and taxonomic annotations, we have added more detail about the location of these data and the availability of related metadata.

“Contigs are available through the JGI genome portal, and taxonomic and functional annotations of these contigs are available on the IMG/M database (<http://img.jgi.doe.gov>), under GOLD project ID Gs0135756. JGI Genome ID’s for each sample, as well as sample metadata, can be found in Chuckran et al (2020; 61).” L210-214

Specific Comments:

L30-31: instead of "anaerobic and autotrophic" don't the authors mean dissimilatory? Assimilatory process of N uptake, fixation, assimilation are upregulated, and dissimilatory processes of nitrification and denitrification were downregulated. This seems to be a clearer distinction than the way it's currently written.

We agree and have changed the wording to: “...dissimilatory transformations of N...” L31

L84: Anoxic conditions (not anaerobic conditions) - environments/conditions are anoxic/suboxic/anoxic, respirations/metabolisms are aerobic/anaerobic

We have changed the wording to read: “...under anoxic conditions...” L91

L95: substrate for ammonia monooxygenase is ammonia so this should be "ammonia is oxidized" not "ammonium is oxidized".

To address this, we have changed the ammonium to ammonia, now on L102.

L94-95: The way this sentence is written, it sounds like this all happens in the same cell. This can be the case if the microbe is performing comammox, but that is probably mostly happening in wastewater treatment plants and not most agricultural soils. Please clarify that nitrification generally is a step-organism process: one ammonia oxidizer and one nitrite oxidizer. Add a citation such as Stein and Klotz 2016 "The Nitrogen Cycle" Curr Biol

We now clarify that nitrification often occurs in a two-step process and occasionally through comammox. The text now also cites additional material, including the reference the reviewer suggested. L102-104 now reads:

“In nitrification, ammonia is oxidized to nitrite and then nitrate. Often the steps of this process occur in different organisms (33), however complete ammonia oxidizers have also been described (34, 35).”

P227: instead of "pre/post fumigation", please write about the effect on microbial biomass,

which is what the fumigation is measuring, rather than the name of the method used to measure it (not all readers will read the methods carefully)

To address this, we have altered the wording of L253-255 to be the following:

“Microbial biomass C (MBC) moderately decreased throughout the incubation (Fig. S2A) and microbial biomass N (MBN) remained constant (Fig. S2B).”

P231: "that changes in microbial C can be attributed to unextracted glucose remaining from the initial K₂SO₄ extraction" - this doesn't make sense to me. Please explain more clearly.

In order to clarify our explanation of these data, we have added additional text and adjusted some of the wording:

“Since the method of microbial biomass extraction used involves two extractions on the same sample (one before and after fumigation), incomplete extraction of the added glucose in the first extraction could result in an artificially high estimate of biomass C. We believe that it is far more likely that microbial biomass and stoichiometry did not change, and that changes in estimated MBC are likely the result of unextracted glucose remaining from the initial K₂SO₄ extraction.” L257-261

L263: add citation here

We have added the following reference for this information, now at L298:

Smith EL, Austen BM, Blumenthal KM, NYC JF. 1975. Glutamate Dehydrogenases Enzymes 3rd ed. Academic Press.

L315-316: "substantial decrease in soluble C" - clarify that this is dissolved inorganic carbon in soil (as opposed to C in biomass, which didn't significantly change)

We have changed the wording to “...K₂SO₄ extractable organic C...”. L357

Further, we have clarified this throughout the manuscript, changing all mention of “soluble C” or “dissolved C” to now, “K₂SO₄ extractable organic C”. We believe that this choice of words better explains the extractant and avoids confusion.

L349: change "assimilation" to "uptake"

We agree with the reviewer that the wording here was confusing and to help clarify, we have changed the wording to the following:

“Similarly, we observed an upregulation of genes associated with nitrate and nitrite transport (KEGG module M00615)...” L389-390

L415: clarify that it was assimilatory nitrate and nitrite reduction

Done, now reads: "...assimilatory nitrate and nitrite reduction..." L477

Supplemental Table S1, S2: gene names should be italicized

Gene names in supplemental table S1 and S2 are now italicized

Figure 1B: Move to supplemental and remove trend line/curve. Too much scatter in data to include that line.

To address this, we removed the scatterplot and trend line, opting instead for a line graph with standard error bars. We respectfully disagree with the reviewer's recommendation to move this figure to the supplement. The figure displays a trend in the concentration of ammonium that we believe, regardless of variability, is important to include in the main text.

Figure 1D: these data are referred to as "dissolved inorganic carbon" in the text and soluble C in the figure. I prefer DIC, but either way, please be consistent. This is easily confused with the mention of soluble C in biomass.

We have changed the text in the figure legend to match that of the main text (K_2SO_4 extractable organic carbon).

Reviewer #2 (Comments for the Author):

This is an interesting study exploring the short-term effects of carbon supplementation on the activity of the soil microbiota. As far as I can judge, the data are sound and support the authors' conclusions. The paper is clearly written and the data nicely presented. I have only a few minor comments.

We thank the reviewer for their time in reviewing our manuscript and appreciate their positive assessment of the work and the writing.

Specific comments

1. Line 116. Is there a particular reason for the choice of glucose as the C supplement?

We chose glucose for two reasons: First, glucose represents a common form of labile carbon which stimulates soil microbial communities. Second, glucose has been widely used when probing the response of soil microbial communities to C limitation. Capturing the transcriptional response of soil microbial communities to glucose is therefore a contribution to this literature, as well as a tried-and-tested method for alleviating C limitation for soil microbes.

“We selected glucose as it is a form of labile C commonly found in soils (51), and has been widely used to alleviate C limitation in soil microbial communities as a means to study growth (52, 53) and metabolic activity (50). L125-127

2. Line 243. ...none of these genes was...

This line has been reworded to “...not one of these genes was directly...” L273

3. Lines 281-282. I think this should be ' ... periplasmic cytochrome c nitrite reductase ...'?

This is correct, Nrf is a nitrite reductase and we have changed the wording to reflect this. L317

4. Line 286. Note that narI and narV encode only one of the subunits of the nitrate reductase (in other words NarV and NarI alone are not active enzymes).

***To address this, we have reworded the section to read:
“with the exception of genes encoding for nitrate reductase (narI/narV; which also functions as a nitrate reductase in DNRA).” L321-322***

5. Lines 289-290. '...did we observed...'. Revise.

Revised to read “...we did observe...” L326

6. Line 348. '...the majority of dissimilatory from t0.' Revise.

We omitted this part of the sentence, formerly at L348. We found that it neither read well nor contributed much to the meaning of the sentence.

7. Have the authors done a chemical analysis of the soil samples used in this study? I am just curious as to whether the availability of other nutrients (iron, for example) might be limiting the growth and/or activity of the soil microbiota.

Unfortunately, we do not have measurements of other nutrients throughout the incubation. We agree that this would provide an interesting opportunity to track colimitation and/or transient limitation.

Reviewer #3:

The manuscript submitted addresses a very important question in soil microbial ecology, searching to identify the linkages that exist between episodic inputs of labile carbon and nitrogen cycling, since only a few studies have looked at this very important connection in terrestrial environments. The authors have used a combination of metagenomics and metatranscriptomics to analyze the shifts caused by the addition of a small amount of glucose to jar incubations of a

mixture of soils. The article is extremely well written, and the questions and some of the answers obtained by this work are very promising, but there seem to be a couple of major methodological issues that need to be addressed:

We thank the reviewer for their time in reviewing our work, and appreciate their positive assessment of the writing.

- There seems to be an overreliance on metagenomics and metatranscriptomics to identify changes in very specific N-cycling and C central metabolism that could have been targeted with qRT-PCR. There should at least be some control experiments that target some of the key genes identified throughout the paper (amtB, narIV, nirBD, nrfAD, nifHDK, glnA, icd) using this technique that narrows down the analysis to confirming what the metatranscriptomic data hints at.

We agree with the reviewer that the addition of qRT-PCR would be a valuable contribution to this study, however we unfortunately do not have these data. We believe that our work still has merit and offers insights into target specific N-cycling genes. Although many of these genes could have been targeted with qRT-PCR, we believe that there were several advantages to a metatranscriptomic approach. First, with increasing utilization of high throughput sequencing, identifying these patterns through metatranscriptomics represents an interesting and useful proof of concept. No study has observed the short-term metatranscriptomic response of soils to labile C inputs, and these results therefore offer a first-look into this response. Second, since metatranscriptomes don't require targeting of specific genes, we were able to detect transcriptional responses for a large number of genes, including many processes that will be described in future publications. Moreover, the metatranscriptomic approach allowed us to assess the response of these N cycling genes in the context of the expression of all other detected transcripts (such as with the SIMPER analysis) and allowed for the comparison against genes responsible for C uptake and glucose breakdown.

- There seems to be an important set of controls missing to address changes in the soil community over the same period of time ****without**** glucose. It seems that a control that only increases the water holding capacity in the same way that the glucose inoculation using only water should be used to keep track of the same gene expression patterns over the same period of time. Since these controls do not seem to have been performed for the metagenomic/metatranscriptomic portion of the work, it seems important to set up an experiment in which jar incubations with and without glucose are used to assess the expression of the genes listed above using a simpler setup (qRT-PCR).

A parallel incubation was conducted where water was added without glucose and we did not find a substantial increase in respiration in response to water alone. These soils were from a mesic environment and we suspect that any type of Birch Effect was therefore relatively small. Ideally, we would have metagenomes and metatranscriptomes of soils amended with just water for each time point, however since metagenomes and metatranscriptomes are costly and difficult to produce, and because we had good evidence that much of the stimulation could be attributed to the addition of glucose, we opted to only sequence samples amended with glucose.

In order to show that we considered the influence of water on microbial activity, we have included details of this parallel incubation. At L165-L169:

“Since the addition of water may stimulate microbial community activity and respiration, especially when starting with very dry soil (56, 57), we measured respiration in a parallel incubation wherein the same volume of water was added without glucose. Headspace CO₂ from these jars was measured and compared against the glucose additions in order to determine the overall effect of glucose and water on microbial respiration.”

And we mention the overall effect of the water additions in the results, as well as reference a new supplemental figure showing these data (Figure S1). L234-235

“We found that the addition of water only slightly influenced CO₂ production (Fig. S1), indicating that the majority of the stimulation was due to the addition of labile C.” L247-248

MINOR ISSUES:

- What about the effect of these levels of N cycling in the long term? From the point of view of lasting impact, it would be useful to know if expression levels and N species are affected long term by this small C input. What happens after a week? A month?

Since our incubation was only for 48 hours and included destructive sampling, we cannot speak to the long-term impacts of this increase in N cycling. However we agree that this is an important question have included at the end of the discussion. L465-467:

“Further, this work focused on a relatively short timeframe, however whether this increase in transcription persists or influences nutrient cycling on the scale of weeks to months remains to be seen.”

- P. 8, l. 140: Why were soils mixed instead of analyzing all three soils separately and get a broader view if this is something that happens in all soil types?

We felt that comparing these soils separately would be of limited benefit since they were fairly similar, and we wanted to have uniform soils for the incubation in order to reduce any variability between jars. We agree that a comparison of multiple soil types would be very interesting, however due to the high cost of sequencing and uncertainty of success we opted to use one soil in this study. Similar to the comment above, we felt this was an important observation and have included in near the end of the discussion. L460-465

“More work needs to be done focusing on this response in a variety soils, as nutrient availability and other soil properties will undoubtedly influence this process. For example, soils high in C and low in N would likely not demonstrate a similar response as observed for this agricultural soil. Understanding how ecosystem properties influence the dynamics of transcriptional profiles is therefore necessary in determining short-term microbial contributions to biogeochemical cycling”

- P. 8, l. 143: It would be useful to know what kind of preliminary work was done to decide on such a specific concentration of glucose

This concentration was chosen based on previous amendment studies and was verified in a trial incubation. We now include details on this selection process on L 157-160

- P.12, l. 222: Are there readings available of the DOC prior to the addition of glucose? It would be useful to have a clear starting point prior to the dynamics for the next 48 hours.

Although we agree with the reviewer that these data would be interesting, we unfortunately do not have them. The K_2SO_4 extractable organic carbon measured for microbial biomass occurred after glucose addition. We did not extract an unamended sample, as it was not the original intention of these data to serve as a measure of available glucose.

- P. 13, l. 243: Expand and explain a little bit of which genes were found here through the metagenomic work. Were there categories of genes that were more abundant? Considering the effort of producing all this sequencing data, it seems like there should be more that can be said here.

We now expand on those gene which were found to drive differences in our metagenomes L274-278. We have also included a separate table to Supplemental Table S3, which now shows the KO numbers found to have the largest LFC between metagenomes. Finally, we have included a brief discussion of these results in L448-452, however we keep the interpretation fairly limited since these proteins are in low abundance and therefore are more likely to demonstrate large LFCs.

“Among these were the subunits of RNA polymerase *rpoB* and *rpoC*, which were in slightly lower abundance at t_8 (LFC -0.1, FDR > 0.1), and the regulatory gene for the *lac* operon, *lacI*, which was in a greater abundance at t_{24} and t_{48} (LFC 0.7, FDR < 0.01). The largest changes were found at t_{24} for low-abundant spore germination proteins (Table S3B), specifically *gerKC* (K06297) and *yfkQ* (K06307) which were 8.8 and 7.4 LFC more abundant than at t_0 .”

- P. 17, l. 324: it should read "microbially mediated"

Fixed to now read “microbially mediated””. L365

- P. 21, l. 411: remove "ex" from reference 72.

“ex” removed, L457

March 11, 2021

Dr. Peter Francis Chuckran
Northern Arizona University
Center for Ecosystem Science and Society
1899 S San Francisco St
Flagstaff, Arizona

Re: mSystems00161-21 (Rapid response of nitrogen cycling gene transcription to labile carbon amendments in a soil microbial community)

Dear Dr. Peter Francis Chuckran:

The reviewers were pleased with your modifications and just have one minor comment remaining. I believe if this is addressed the manuscript will be ready for publication. Below you will find the comments of the reviewers.

To submit your modified manuscript, log onto the eJP submission site at <https://msystems.msubmit.net/cgi-bin/main.plex>. If you cannot remember your password, click the "Can't remember your password?" link and follow the instructions on the screen. Go to Author Tasks and click the appropriate manuscript title to begin the resubmission process. The information that you entered when you first submitted the paper will be displayed. Please update the information as necessary. Provide (1) point-by-point responses to the issues raised by the reviewers as file type "Response to Reviewers," not in your cover letter, and (2) a PDF file that indicates the changes from the original submission (by highlighting or underlining the changes) as file type "Marked Up Manuscript - For Review Only."

Due to the SARS-CoV-2 pandemic, our typical 60 day deadline for revisions will not be applied. I hope that you will be able to submit a revised manuscript soon, but want to reassure you that the journal will be flexible in terms of timing, particularly if experimental revisions are needed. When you are ready to resubmit, please know that our staff and Editors are working remotely and handling submissions without delay. If you do not wish to modify the manuscript and prefer to submit it to another journal, please notify me of your decision immediately so that the manuscript may be formally withdrawn from consideration by mSystems.

Sincerely,

Ryan McClure

Editor, mSystems

Journals Department
Reviewer comments:

Reviewer #2 (Comments for the Author):

I have just one remaining change to suggest.

"genes encoding for the nitrate reductase (*narI/narV*; which also functions as a nitrate reductase in DNRA).

This statement is still confusing. The respiratory nitrate reductase is a three subunit enzyme encoded by a four gene operon (one gene product has a role in maturation). In *E. coli*, there are two nitrate reductases (sometimes called NRA and NRZ) and *narI* and *narV* encode homologous subunits of those two enzymes, the two operons are *narGHJI* and *narZYWV*. It does not make sense to say that *narI/narV* encode nitrate reductase.

For the same reason, elevated nitrate reductase activity is expected only if the entire operon is up-regulated. This appears not to be the case in the authors' data since they mention only up-regulation of *narV* and *narI*.

We thank the reviewers again for their assessment of our work, and hope the following changes sufficiently address their comment. Please find our responses in *italicized bold* font below each reviewer comment. The line numbers of each change correspond to the “Marked Up Manuscript Final” file.

Reviewer #2 (Comments for the Author):

I have just one remaining change to suggest.

"genes encoding for the nitrate reductase (narI/narV; which also functions as a nitrate reductase in DNRA).

This statement is still confusing. The respiratory nitrate reductase is a three subunit enzyme encoded by a four gene operon (one gene product has a role in maturation). In E. coli, there are two nitrate reductases (sometimes called NRA and NRZ) and narI and narV encode homologous subunits of those two enzymes, the two operons are narGHJI and narZYWV. It does not make sense to say that narI/narV encode nitrate reductase.

For the same reason, elevated nitrate reductase activity is expected only if the entire operon is up-regulated. This appears not to be the case in the authors' data since they mention only up-regulation of narV and narI.

To clarify, we have changed the wording at 316-318 to the following:

“Similarly, expression for most denitrification genes were downregulated throughout the incubation, with the exception of narI and narV, which encode for gamma subunits of nitrate reductase.”

We have also altered the wording at 307 to read:

“Nitrate reductase subunits (narI/narV) were upregulated...”

To convey that these genes encode for nitrate reductase subunits.

March 30, 2021

Dr. Peter Francis Chuckran
Northern Arizona University
Center for Ecosystem Science and Society
1899 S San Francisco St
Flagstaff, Arizona

Re: mSystems00161-21R1 (Rapid response of nitrogen cycling gene transcription to labile carbon amendments in a soil microbial community)

Dear Dr. Peter Francis Chuckran:

Your manuscript has been accepted, and I am forwarding it to the ASM Journals Department for publication. For your reference, ASM Journals' address is given below. Before it can be scheduled for publication, your manuscript will be checked by the mSystems senior production editor, Ellie Ghatineh, to make sure that all elements meet the technical requirements for publication. She will contact you if anything needs to be revised before copyediting and production can begin. Otherwise, you will be notified when your proofs are ready to be viewed.

- Minimum resolution of 1280 x 720
- .mov or .mp4. video format
- Provide video in the highest quality possible, but do not exceed 1080p
- Provide a still/profile picture that is 640 (w) x 720 (h) max

We recognize that the video files can become quite large, and so to avoid quality loss ASM suggests sending the video file via <https://www.wetransfer.com/>. When you have a final version of